# Fabrication of Large-Area Nanostructures Using Cross-Nanoimprint Strategy

**DOI:** 10.3390/nano14120998

**Published:** 2024-06-08

**Authors:** Yujie Zhan, Liangui Deng, Wei Dai, Yongxue Qiu, Shicheng Sun, Dizhi Sun, Bowen Hu, Jianguo Guan

**Affiliations:** 1State Key Laboratory of Advanced Technology for Materials Synthesis and Processing, School of Materials Science and Engineering, Wuhan University of Technology, Wuhan 430070, China; zhanyujie@whut.edu.cn (Y.Z.); yxqiu1999@whut.edu.cn (Y.Q.); scheng_sun@whut.edu.cn (S.S.); dizhis@whut.edu.cn (D.S.); bowenhu@whut.edu.cn (B.H.); 2School of Information Engineering, Wuhan University of Technology, Wuhan 430070, China; 3Suzhou Institute, Wuhan University, Suzhou 215028, China; 4Center for Nanoscience and Nanotechnology, Key Laboratory of Artificial Micro- and Nano-Structures of Ministry of Education, School of Physics and Technology, Wuhan University, Wuhan 430072, China; weidai@whu.edu.cn

**Keywords:** nanoimprint lithography, composite mold, large-area nanostructures

## Abstract

Nanostructures with sufficiently large areas are necessary for the development of practical devices. Current efforts to fabricate large-area nanostructures using step-and-repeat nanoimprint lithography, however, result in either wide seams or low efficiency due to ultraviolet light leakage and the overflow of imprint resin. In this study, we propose an efficient method for large-area nanostructure fabrication using step-and-repeat nanoimprint lithography with a composite mold. The composite mold consists of a quartz support layer, a soft polydimethylsiloxane buffer layer, and multiple intermediate polymer stamps arranged in a cross pattern. The distance between the adjacent stamp pattern areas is equal to the width of the pattern area. This design combines the high imprinting precision of hard molds with the uniform large-area imprinting offered by soft molds. In this experiment, we utilized a composite mold consisting of three sub-molds combined with a cross-nanoimprint strategy to create large-area nanostructures measuring 5 mm × 30 mm on a silicon substrate, with the minimum linewidth of the structure being 100 nm. Compared with traditional step-and-flash nanoimprint lithography, the present method enhances manufacturing efficiency and generates large-area patterns with seam errors only at the micron level. This research could help advance micro–nano optics, flexible electronics, optical communication, and biomedicine studies.

## 1. Introduction

In recent years, with the continuous expansion of micro–nanofabrication application fields and the ongoing development of micro–nano optics and photonics, there has been a growing demand for large-area, high-precision micro–nanostructured devices such as optical waveguides [1,2], OLEDs [3], multifunctional image displays [4,5], photodetectors [6], biosensors [7], metalenses [8,9], and solar cells [10]. The high-precision manufacturing of these devices over large areas is a key challenge for the practical implementation of such technologies. Historically, photolithography has dominated the field of micro–nanofabrication. While techniques such as electron beam lithography [11], X-ray lithography [12], extreme ultraviolet lithography [13], and ion beam lithography [14] can achieve ultra-high resolution, they suffer from low throughput and high processing costs. Interference lithography [15], laser direct writing [16], and molecular self-assembly [17] techniques offer the efficient and cost-effective fabrication of large-area nanostructures. Nevertheless, meeting the requirements for high-precision micro- and nanostructure preparation poses a significant challenge for such techniques. Unlike other lithography techniques, nanoimprint lithography (NIL) transfers patterns from a template onto a substrate coated with a large-area imprint-resistant material using heat or ultraviolet (UV) light while applying constant pressure, thereby enabling the mass production of highly precise nanostructures [18,19], such as optical waveguides or other photonic integrated devices with precision and low losses [20,21].

As the efficiency of NIL and the manufacturing capabilities of templates continue to expand, various large-area nanoimprint techniques have been developed. There are three main approaches: full wafer NIL [22,23,24], roll-type NIL [25,26,27], and step-and-repeat NIL [28,29]. Because nanoimprint lithography is a one-to-one replication process, challenges arise in the fabrication of large-area nanoimprint molds for both full wafer NIL and roll-type NIL. Step-and-repeat NIL involves imprinting in a step-by-step manner using smaller-sized patterned templates to obtain larger patterned areas. Therefore, this method is commonly employed in the fabrication of large-area, high-precision nanostructures. However, the leakage of UV light and overflow of imprint resin will impact the subsequent imprinting [30,31], leading to obvious seams that ultimately affect the functionality and efficiency of the resulting devices. To address these challenges, Lee, J et al. addressed UV light leakage by applying a chromium layer to non-patterned mold areas, using collimated UV light [32]. However, UV light diffraction led to premature resin curing in subsequent areas. Other research has applied overlapping imprinting to achieve seamless large-area nanopatterns [33,34,35,36], but such attempts resulted in defects at overlapping regions, affecting device performance. Kataza Shingo et al. proposed a double lithography method enabling large-scale continuous nanostructure fabrication while sacrificing efficiency [37]. In conclusion, achieving seam accuracy while maintaining efficiency remains a challenge in step-and-repeat NIL for large-area nanostructure fabrication.

In this paper, we propose an efficient method for fabricating large-scale nanostructures with small seams. This method involves preparing a composite mold and utilizing a cross-nanoimprint strategy. By arranging three identical intermediate polymer stamps (IPSs) in a parallel and equidistant cross pattern on a quartz substrate coated with soft polydimethylsiloxane (PDMS), this composite mold combines the high-precision imprinting of hard molds with the uniform large-area imprinting capability of soft molds, thereby enhancing the efficiency of imprinting. Employing primary nanoimprinting and etching followed by secondary cross-nanoimprinting and etching significantly reduces the overflow of imprint resin and UV light leakage. Scanning electron microscopy (SEM) results demonstrated that the dimensions of the fabricated seams were in the micrometer range. For users with requirements for large-area high-performance devices with periodicity, such as large-area metalenses, photonic waveguides, and flat panel displays, this method represents a novel approach for the low-cost and efficient fabrication of large-scale nanostructures with high seam accuracy.

## 2. Materials and Methods

### 2.1. Fabrication Strategy

The overall fabrication strategy is illustrated in Figure 1. The process primarily involves preparing a composite mold and using a cross-nanoimprint approach to fabricate large-area nanostructures. Initially, we assume *m* to be the number of IPSs on the composite mold (*m* represents positive integers, with *m* greater than 1), and *T* denotes the side length of the square nanopattern on original mold 1. Then, the mass-replication of IPSs is performed using original mold 1 with dimensions of *T* × *T*, which features square nanoholes, as depicted in Figure 1a. Subsequently, composite mold 1 is prepared using a counterpoint system and a displacement rotating platform. IPSs are arranged in parallel and intersected on a quartz substrate coated with soft PDMS, based on a repetitive lateral pattern and variable longitudinal design. Based on a side view of composite mold 1, the quartz substrate serves as a support layer with a high elastic modulus (72–84 GPa) and offers excellent UV transparency to prevent lateral deformation during imprinting. Soft PDMS functions as a cushioning layer with a lower elastic modulus (2 MPa) and excellent UV transparency. These properties ensure tight conformal contact between non-flat substrates and molds, making it suitable for large-area nanoimprinting. IPSs, which comprise the imprint layer, are a type of transparent fluoropolymer with a higher elastic modulus (4000 MPa) and lower surface energy than polyethylene terephthalate (PET, 2000 MPa). Therefore, IPSs offer excellent mechanical properties for high-precision nanostructure imprinting. The top view of composite mold 1 shows that *m* IPSs with dimensions of *T* × *T*, featuring square nano-pillars, are arranged in parallel with a lateral spacing of *T*, facilitated via the micron-level alignment markers on the quartz substrate and IPSs.

Subsequently, as depicted in Figure 1b,c, the pattern is first transferred to the silicon substrate using composite mold 1 through the first nanoimprinting and etching process. Subsequently, after the lateral displacement of *T*, the second cross-nanoimprinting and etching process is employed to complete the fabrication of large-area nanopatterns measuring *T* × 2*mT* and featuring square nanoholes, as shown in Figure 1d. To continue obtaining larger area nanopatterns, we can treat this mold as original mold 2 and repeat the steps in Figure 1a–c. This iterative process could achieve larger area nanopatterns with dimensions of 2*mT* × 2*mT* (original mold 3). Similarly, further increasing the number of step iterations for imprinting and the quantity of IPSs on the composite mold could expand both the horizontal and vertical dimensions.

We next compare the efficiency of the proposed fabrication strategy with that of traditional step-and-repeat nanoimprint lithography, focusing solely on the nanoimprinting steps (excluding the time spent on alignment, etching, etc.). We assume *n* to be the number of times the nanoimprinting is performed in a crossover manner (in Figure 1b,c, *n* equals one, indicating two imprints). Then, *t* represents the time required for one imprint. The area of the nanostructures obtained from the imprinting is denoted as *S*(*m*,*n*), with the following equation:*S*(*m,n*) = (2*m*)*^n^ T*^2^.(1)

The time required to prepare nanostructures with the same area *S*(*m*,*n*) using the conventional step-and-repeat nanoimprinting lithography and our fabrication strategy is denoted as *t*_1_ and *t*_2_, respectively. The ratio between these values is denoted as *η* and uses the following equation [38]:(2)$$\eta =\frac{t_{1}}{t_{2}}=\frac{{\left[(2m\right)}^{n}+1]t}{(m+2)\;nt}=\frac{{(2m)}^{n}+1}{(m+2)\;n}.$$

Equation (2) demonstrates that when imprinting the same area, the time required by traditional step-and-repeat nanoimprint lithography grows exponentially. As *m* and *n* increase, the efficiency becomes progressively lower compared to that under our approach. For instance, when *n* = 2, and *m* = 3, *η* is 37/10, and our strategy achieves a 270% improvement in imprinting efficiency compared to that under traditional step-and-repeat nanoimprint lithography. For the ease of calculation, if *T* is 5 mm, and *t* is 3 min, and we aim to prepare nanopatterns at the meter scale, we assume a size of 1280 mm × 1280 mm; according to Equation (1), *m* is 8, and *n* is 4. The time required for the traditional step-and-repeat nanoimprint lithography is approximately 3276 h, while our method only requires 2 h. Thus, the ratio between these two approaches is approximately 1638. Therefore, our proposed approach of using a composite mold combined with cross-nanoimprinting could significantly enhance the efficiency of preparing large-area nanopatterns. In this study, *m* is 3, *n* is 1, and *T* is 5 mm, thereby achieving large-area nanostructures with dimensions of 5 mm × 30 mm.

### 2.2. The Composite Mold Fabrication Process

Initially, electron beam lithography (EBL) was used in conjunction with the lift-off process and inductively coupled plasma etching (ICP) techniques to create an original mold measuring 5 mm × 5 mm on a two-inch diameter silicon substrate. The specific procedure was as follows. First, we designed the structure. Based on the transmission phase-type metasurface, control over electromagnetic waves can be achieved through manipulating the dimensions of the structural units. This technology has wide potential applications in optical imaging, communications, radar, and other fields. Under this background, we designed metalenses focusing on the visible light spectrum. By design, a positive electron beam resist (PMMA-950k, MicroChem, Newton, MA, USA) was used along with an electron beam lithography system (eLine Plus, Raith, Dortmund, NRW, Germany) to define square nanoholes characterized by periodic repetition in the lateral direction and variation in the longitudinal direction. To create the hard mask, approximately 30 nm of chromium metal was subsequently evaporated on the mold using a thermal evaporator (JSD-400, Ahjiashuo, Hefei, AH, China) and then lifted off in an acetone solvent at 80 °C. Finally, the pattern was transferred to a silicon substrate via plasma etching in C_4_F_8_ and SF_6_ gases. The residual chromium mask was removed using a chrome etchant, thus completing fabrication of the original mold. As a result, as depicted in Figure 2, we achieved a nanopattern of 5 mm × 5 mm on a two-inch diameter silicon substrate. The unit structure consisted of square nanoholes with approximate depths of 160 nm, which varied in side length from 109 nm to 200 nm with a periodicity of 302 nm.

Then, as depicted in Figure 3a, the original mold was used to replicate the IPSs using an Eitre 6-in. Nanoimprinter (Obducat AB, Lund, Sweden). The maximum applied pressure was 50 bar, with a temperature of 160 °C and an imprinting time of 180 s. After the mass replication of IPSs (the original mold can replicate IPSs thousands of times), we began fabrication of the composite mold. As shown in Figure 3b, the quartz substrate underwent surface treatment via immersion in acetone, deionized water, and ethanol solutions, each subjected to 2 min of ultrasonication at 100% power. Subsequently, oxygen plasma was used to increase the surface energy, thereby enhancing the surface’s adhesion to the photoresist. After that, using pre-prepared templates, micron-scale alignment patterns suitable for subsequent alignment operations and testing were created on a quartz substrate coated with an S1805 photoresist through ultraviolet photolithography (H94-37, Nanguang vacuum, Chengdu, SC, China). Following this process, chromium markers were prepared on the surface of the quartz substrate using a lift-off process. Successively, the soft PDMS (Sylgard 184 PDMS, Dow Corning, Midland, MI, USA) prepolymer was mixed with a curing agent (a mixture of vinyl-terminated dimethylsiloxane and organosilicon compounds) in a 10:1 ratio, degassed under a vacuum for one hour, poured onto the quartz substrate surface, spun at 2000 rpm for 60 s, and slowly cured at 60 °C on a hot plate for 12–14 h. Then, using a constructed counterpoint system and a displacement rotating platform, the prepared IPSs were strategically positioned on the surface of the soft PDMS at 5 mm intervals using alignment markers.

As shown in Figure 3c, the fabricated IPSs were arranged on the soft PDMS surface with a spacing of 5 mm using the established counterpoint system and displacement rotating platform, which consists of two components: an optical microscope and a displacement rotating platform. The optical microscope has a resolution of 2 µm, while the displacement platform achieves a positioning resolution of 0.01 mm along the X, Y, and Z axes. This platform also offers a rotation angle accuracy of ±2 arcminutes in the X–Y plane. We achieved alignment through a semi-automatic method. The quartz spin-coated with soft PDMS was affixed onto the substrate; then, we secured the IPSs using a jig. Subsequently, displacement and rotation of the platform were manipulated using an optical system to align the IPSs with the markers. Utilizing this method, we successfully spaced three IPSs at approximately 5 mm intervals on a 35 mm × 35 mm quartz substrate, as depicted in Figure 3d. The IPSs and soft PDMS were connected using a double-sided adhesive tape primarily made of polyethylene terephthalate (PET) substrate and acrylic adhesive. PET substrate offers excellent transmission at 365 nm ± 1 nm, while acrylic adhesives are commonly utilized as a primary component in photosensitive resins owing to their exceptional transparency under UV light, surpassing a transmission rate of 90%. This tape has a smooth surface and a thickness of 50 μm. The pattern size on the IPSs was 5 mm × 5 mm, with a unit structure consisting of square nano-pillars. The pillars had an approximate height of 160 nm, varying in side length from 116 nm to 203 nm, with a periodicity of 310 nm. The imprinting demonstrated good consistency, successfully achieving replication of the IPSs. The fabricated IPSs exhibited complementary pattern structures to those on the original silicon mold.

## 3. Results and Discussion

After fabricating the composite mold, we proceeded to create large-area nanostructures using the two-step nanoimprinting and etching process. As shown in Figure 4a, the first nanoimprinting and etching process began with cleaning the silicon substrate, followed by surface modification using oxygen plasma. The parameters for the plasma treatment were a power of 200 W, an oxygen flow rate of 200 sccm, and a duration of 600 s to increase surface energy. Next, TU7-120k imprint resist (compatible with the IPS process) was spin-coated onto the substrate at 2000 rpm for 60 s and baked at 95 °C for 90 s. Before UV nanoimprinting, it was necessary to apply an anti-sticking treatment to the composite mold. The primary purpose of pre-treating the mold surface was to reduce its surface energy, thereby facilitating separation between the mold and the substrate during the demolding process. In this study, we utilized a long-chain silane, 1H, 1H, 2H, and 2H-perfluorodecyltrichlorosilane (FDTS) for mold anti-sticking treatment. FDTS forms a compact monolayer film by binding to hydrogen bonds on the substrate surface. The specific procedure involved placing the composite mold and FDTS in a high-temperature oven, evacuating the chamber, and then subjecting the samples to 30 min deposition at 80 °C. Experimental results confirmed that the FDTS-treated mold exhibited strong hydrophobicity, making it highly suitable for subsequent imprinting experiments (one IPS can transfer patterns onto the imprint resin approximately ten times). Subsequently, we employed a gradual pressure increase approach to ensure that the composite working molds were completely filled with the imprint resist, with a maximum pressure of 30 bar, an imprinting temperature of 65 °C, and an exposure time of 300 s (LED UV lamp with an emission spectrum at 365 nm ± 1 nm). After imprinting, the mold was demolded, and the residual TU7 imprint resist was removed using oxygen plasma etching. The etching parameters for the reactive-ion etching (RIE) process were as follows: a power of 50 W, a working pressure of 80 mTorr, an oxygen flow rate of 50 sccm, and an etching time of 30 s. Then, the remaining TU7 imprint resist was used as a mask to etch the silicon substrate surface using a reactive ion etching (RIE) system. The etching parameters for the RIE process included a mixture of CHF_3_, SF_6_, and O_2_ gases with flow rates of 300 sccm, 250 sccm, and 95 sccm, respectively; an etching power of 200 W; and an etching time of 120 s, resulting in an etch depth of approximately 300 nm. Finally, the remaining TU7-120k imprint resist was removed using oxygen plasma etching. The nanoimprinting and etching results, as shown in Figure 4b, produced three nanopatterns, each measuring 5 mm × 5 mm, on a silicon substrate, with approximately 5 mm spacing between them. Scanning electron microscopy was used to measure the nanostructures on various patterns, showing excellent consistency in nanoimprinting attributed to the utilization of air pressure.

Figure 5 shows SEM images of the sidewall structure of the TU7 imprint resist after demolding with a composite mold, as well as SEM images depicting the etching transfer onto the silicon substrate. The results after imprinting and etching demonstrate a well-defined pattern morphology.

A second cross-nanoimprinting and etching process was then conducted, as illustrated in Figure 6a, following a similar procedure to the first nanoimprinting and etching process. Initially, oxygen plasma treatment was applied to the first imprint and etching pattern to enhance the surface energy, followed by spin-coating with a TU7-120k imprint resist. Subsequently, alignment was achieved, and UV nanoimprinting was performed using the same parameters. After demolding, reactive ion etching was conducted to remove the residual imprint resist and etch the silicon surface. Finally, oxygen plasma was used to remove the remaining TU7-120k imprint resist.

Ultimately, we employed a two-step nanoimprinting and etching method to fabricate a large-area silicon nanopattern, as shown in Figure 6b, with dimensions of 5 mm × 30 mm (comprising six patterns of 5 mm × 5 mm each). This result confirmed that a large-area patterning process was possible. The estimated cost for one imprint procedure is around USD 14. The final product, measuring 5 mm x 30 mm, costs approximately USD 646.

Figure 6c,d depict the shapes of the patterns transferred onto the silicon substrate after the first and second nanoimprinting and etching processes, respectively. The unit structure consists of square nanoholes with an approximate depth of 300 nm, varying in side length from 105 nm to 210 nm, with a periodicity of 306 nm. A comparison with the nanostructure of the original mold (Figure 2) confirms the presence of minimal or no difference in the linewidths and shapes of the patterns, demonstrating a uniform transfer of the nanostructures. This result is primarily attributed to the use of a composite mold and pneumatic pressure in the imprinting process, ensuring excellent uniformity in imprinting. We also employed a pre-curing method, whereby the imprint resin in the non-patterned areas of the composite mold was pre-cured based on its positioning before nanoimprinting. This pre-curing enhanced the resin’s strength, eliminated defects arising from edge nanoimprinting and etching on the composite mold’s IPS edge, and mitigated the resin’s impact on the second-step cross-imprinting lithography.

Subsequently, we measured the seams at the edge and between the patterns transferred onto the silicon substrate after the first and second nanoimprinting and etching examples using scanning electron microscopy, as shown in Figure 6e,f. The lateral distance between the two patterns was found to be about 15 μm, while the longitudinal distance was about 1 μm. The experimental results indicate that using the composite mold for cross-nanoimprinting and etching significantly reduced the overflow of imprinting resin and the leakage of ultraviolet light, leading to smaller seams between the large-area patterns being fabricated. Nonetheless, the presence of seams persisted due to errors generated during preparation of the composite mold and discrepancies arising from the identical alignment process used during the second round of cross-nanoimprinting and etching. The positions of the micro-scale markers on the IPSs and quartz substrate were measured using an optical microscope, with both lateral and longitudinal alignment errors controlled at the micron level. The margin of error was influenced by the system’s inherent inaccuracies and the non-contact alignment method, which created a gap between the IPSs and substrate. To eliminate this gap, repeated adjustments to the alignment markers were required. However, due to limitations in the optical and sensor systems, some degree of error is possible. Achieving greater precision would require more expensive and complex software and equipment. Employing a Fourier spectrum analysis of the Moiré patterns formed by overlapping identical motifs enabled the alignment needed to create large-scale nanostructures with quasi-seamless features, without alignment marks [39].

## 4. Conclusions

Based on the proposed strategy of using a composite mold and cross-nanoimprint, this study successfully achieved large-area, high-precision nanostructures measuring 5 mm × 30 mm using only a single 5 mm × 5 mm original mold. Subsequent scaling in both the lateral and longitudinal directions could be easily achieved by increasing the number of step-and-repeat NIL cycles and the quantity of IPSs on the composite mold. 

The micrometer-level seam errors primarily resulted from the manufacturing precision of the equipment used. Utilizing high-precision alignment instruments and techniques could further enhance the accuracy of seam alignment. 

Furthermore, the fabricated large-scale nanopatterns could serve as templates for the preparation of full wafer NIL molds, thereby addressing the challenge of fabricating large-area molds. This application demonstrates the scalability and versatility of the proposed method, offering an efficient solution for producing imprint molds for wafer-scale patterning processes.

In summary, the proposed approach significantly enhances the efficiency of large-area nanostructure manufacturing and the accuracy of seams. Our method enables the cost-effective, large-scale production of high-performance devices such as photonic crystals, metalenses, solar cells, photodetectors, and sensors.

## Figures and Tables

**Figure 1 nanomaterials-14-00998-f001:**
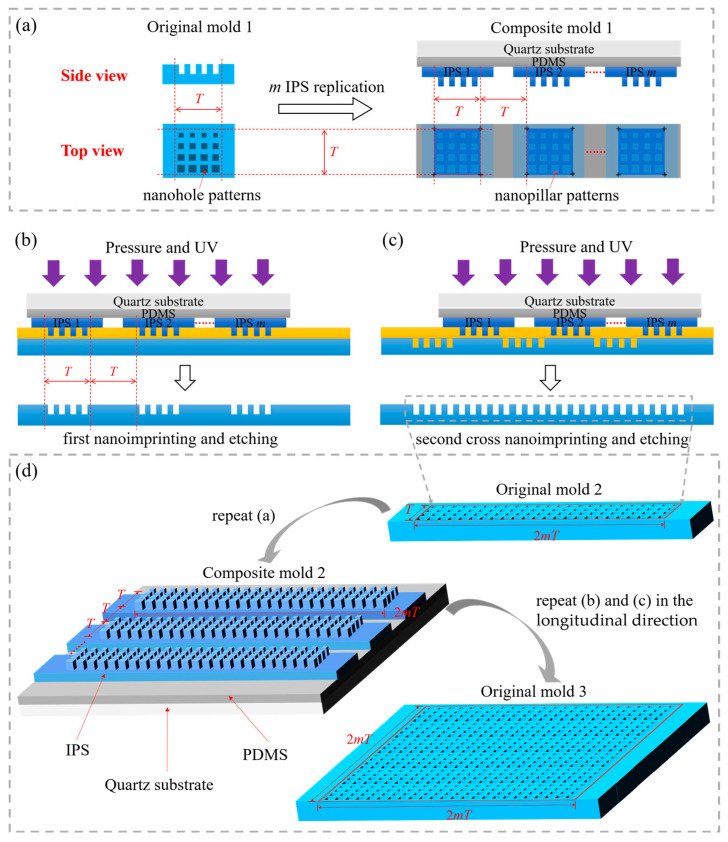
(**a**) Side-view and top-view of original mold 1 and composite mold 1; the cross-nanoimprint strategy with composite mold 1 in the lateral direction, which presents (**b**) the first nanoimprinting and etching results; and (**c**) the second cross-nanoimprinting and etching results; (**d**) the fabrication strategy of larger area nanostructures.

**Figure 2 nanomaterials-14-00998-f002:**

Optical image of the original mold, along with corresponding scanning electron microscopy images of various regions.

**Figure 3 nanomaterials-14-00998-f003:**
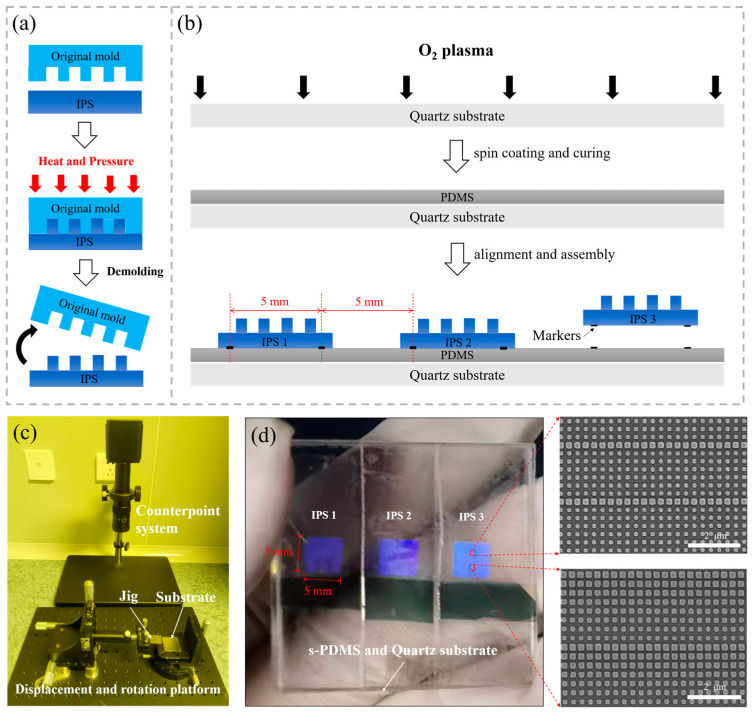
Scheme of the (**a**) IPSs and (**b**) composite mold fabrication process; (**c**) counterpoint system and displacement rotating platform; (**d**) optical image of the composite mold with three IPSs, along with corresponding scanning electron microscopy images of various regions.

**Figure 4 nanomaterials-14-00998-f004:**
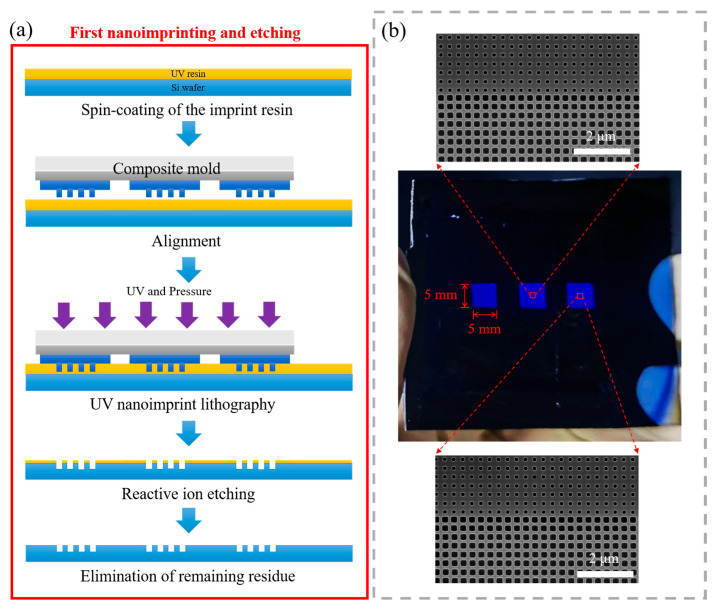
(**a**) Scheme of the first nanoimprinting and etching process; (**b**) optical image of the nanoimprinted silicon pattern, along with corresponding scanning electron microscopy images of various regions.

**Figure 5 nanomaterials-14-00998-f005:**
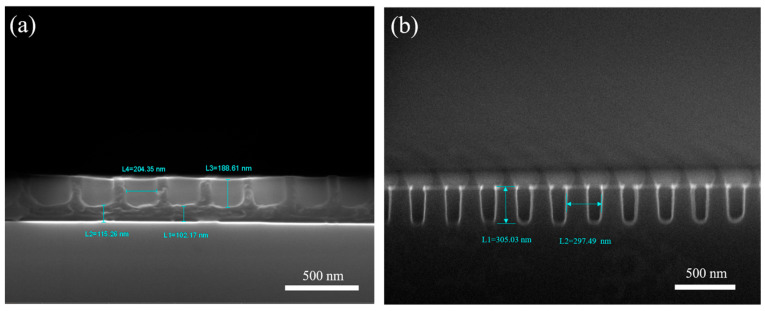
SEM images of the sidewall structure of the (**a**) TU7 imprint resist and the (**b**) silicon substrate (the residual TU7 imprint resist remaining on the substrate).

**Figure 6 nanomaterials-14-00998-f006:**
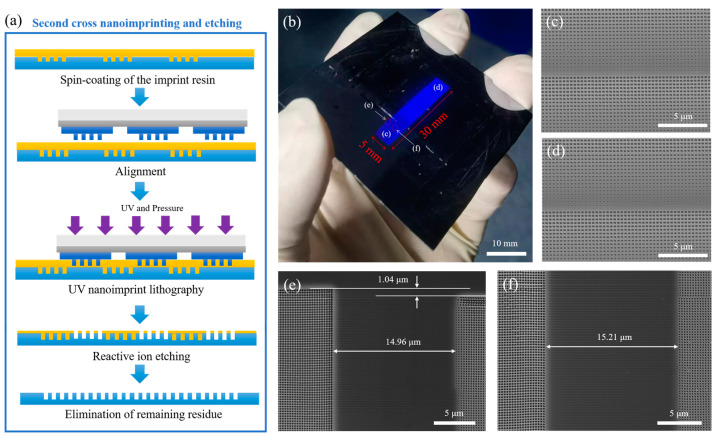
(**a**) Scheme of the second cross-nanoimprinting and etching process; (**b**) large-area nanoimprinted silicon pattern measuring 5 mm × 30 mm; (**c**,**d**) the results of the hole pattern transferred onto the silicon substrate after the first and second nanoimprinting and etching processes, respectively; (**e**,**f**) the seams at the edge and in the middle, respectively, between the patterns of the first and the second nanoimprinting and etching examples.

## Data Availability

The data presented in this study are available on request from the corresponding author.

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
