# Peer review of "Fabrication of Large-Area Nanostructures Using Cross-Nanoimprint Strategy"

_nanomaterials, 2024, doi:10.3390/nano14120998_

Round 1
Reviewer 1 Report
Comments and Suggestions for Authors
The manuscript “Fabrication of large-area nanostructures by cross nanoimprint strategy” by Yujie Zhan, Liangui Deng, Wei Dai, Yongxue Qiu, Shicheng Sun, Dizhi Sun, Bowen Hu and Jianguo Guan proposes an efficient method for fabrication of large-area nanostructure using step-and-repeat nanoimprint lithography with a composite mold. The method proposed allows obtaining large-area patterns with a seam error only at the micron level, and contributes to micro-nano optics, flexible electronics, optical communication, and biomedicine areas.
The manuscript is clearly written, well-organized and the topic is of considerable interest for the research community.
In reviewer´s opinion, it is probably better to use term “manufacturing” instead of “fabrication” or “fabricating” in the manuscript text.
What is the origin of Equations (1) and (2)? Literature reference should be added.
What sizes of individual nanoholes and nanopillars of original molds? How were respective sizes chosen?
The authors claimed that their method leads to 270% improvement in imprinting efficiency when compared with traditional step-and-repeat nanoimprint lithography. How this percentage was evaluated?
In Conclusions section the reference [36] in “Employing Fourier spectrum analysis of Moiré patterns formed by overlapping identical motifs enables the alignment necessary for creating large-scale nanostructures with quasi-seamless features and without alignment marks [36]”. According to the reviewer´s opinion, this detail should be mentioned and discussed in other section of the manuscript.
In the Conclusions the authors may use bullet structure in order to highlight better importance and novelty of the study.
Reviewer 2 Report
Comments and Suggestions for Authors
The authors have presented an interesting work on large-area fabrication of nanostructure by nanoimprint method. I have the following suggestions to enhance the quality of the paper.
1) Apart from NIL, other coating methods should also be briefly discussed in the Introduction section such as: https://doi.org/10.3390/coatings12081115.
2) The author didn't comment on the quality of the imprinted structures. Is it possible to provide magnified images taken by SEM to show the side walls and the spacing between the structures? Because, in NIL, usually the resolution and quality are limited.
3) Is fabricating optical waveguides or other photonic integrated devices with precision and low losses possible?
4) Did the author fabricate any meaningful optical element such as metalens and characterize it to determine its performance? It will be good to show some characterization results.
5) What is the cost of each imprint procedure? How much can a final product of size 5mm x5 mm cost?
6) How precise is the alignment of the original stamp and the substrate? And how do you control it? Manually or automatically?
7) How many times, one mold can be used? Can stamps deteriorate with time and usage?
Comments on the Quality of English Language
None
Reviewer 3 Report
Comments and Suggestions for Authors
This is a useful manuscript for the readers. I have some comments and I'd like to see the answers to these comments in the revised manuscript.
Is the suggested cross nanoimprint method better for the user than other listed methods? The Introduction Section which looks like a literature review lists other methods. Or it is a possible new approach that may be suitable for some users. The method seems rather complex.
Most of the numerical values are presented without their determination errors and look like world constants.
Line 115. What is the Young's modulus of the used PET?
Line 250. Oxygen is O. The authors mean dioxygen O2.
I've never seen an abbreviation s-PDMS. Usually, PDMS may be called soft.
Line 210. PET film has excellent transparency when we look with the naked eye. PET absorbs light with wavelengths lower than 300-320 nm. Moreover, the used PET was covered with double-sided adhesive (line 209). What was the absorption (or transmittance) of such film? The tape was made primarily (?) of PET.
What UV lamp was used? What is its emission spectrum?
Line 193. What is the curing agent? Sn-compound?
Comments on the Quality of English Language
Please verify once more.
Round 2
Reviewer 2 Report
Comments and Suggestions for Authors
I am willing to accept the paper in its current form
Comments on the Quality of English Language
None.
